# Ablation of Acid Ceramidase Impairs Autophagy and Mitochondria Activity in Melanoma Cells

**DOI:** 10.3390/ijms22063247

**Published:** 2021-03-23

**Authors:** Michele Lai, Veronica La Rocca, Rachele Amato, Giulia Freer, Mario Costa, Pietro Giorgio Spezia, Paola Quaranta, Giuseppe Lombardo, Daniele Piomelli, Mauro Pistello

**Affiliations:** 1Retrovirus Center and Virology Section, Department of Translational Research and New Technologies in Medicine and Surgery, University of Pisa, 56100 Pisa, Italy; vero.laro@gmail.com (V.L.R.); giulia.freer@unipi.it (G.F.); piergiorgiospezia@gmail.com (P.G.S.); paola.quaranta@unipi.it (P.Q.); giuseppe.lombardo93@libero.it (G.L.); mauro.pistello@unipi.it (M.P.); 2Sant’Anna School of Advanced Studies, 56100 Pisa, Italy; rachele.agatamato@gmail.com; 3Institute of Neuroscience, Italian National Research Council (CNR), 56100 Pisa, Italy; mario.costa@in.cnr.it; 4Scuola Normale Superiore, 56100 Pisa, Italy; 5Anatomy and Neurobiology, University of California, Irvine, California, 829 Health Sciences Rd, Irvine, CA 92617, USA; Piomelli@hs.uci.edu; 6Virology Unit, Pisa University Hospital, 56100 Pisa, Italy

**Keywords:** acid ceramidase, adjuvant therapy, melanoma, autophagy, cancer biology, ceramides

## Abstract

Cutaneous melanoma is often resistant to therapy due to its high plasticity, as well as its ability to metabolise chemotherapeutic drugs. Sphingolipid signalling plays a pivotal role in its progression and metastasis. One of the ways melanoma alters sphingolipid rheostat is via over-expression of lysosomal acid ceramidase (AC), which catalyses the hydrolysis of pro-apoptotic long-chain ceramides into sphingosine and fatty acid. In this report, we examine the role of acid ceramidase in maintaining cellular homeostasis through the regulation of autophagy and mitochondrial activity in melanoma cell lines. We show that under baseline conditions, wild-type melanoma cells had 3-fold higher levels of the autophagy marker, microtubule-associated proteins 1A/1B light chain 3B (LC3 II), compared to AC-null cells. This difference was further magnified after cell starvation. Moreover, we noticed autophagy impairment in A375 AC-null cells, possibly due to local accumulation of non-metabolized ceramides. Nonetheless, we observed that AC-null cells exhibited a significant increase in mitochondrial membrane potential compared to control cells. Consistent with this observation, we found that, after total starvation, ~30% of AC-null cells undergo apoptosis compared to ~6% of wild-type cells. As expected, AC transfection restored viability in A375 AC-null cells. Together, these findings suggest that AC-null melanoma cells change and adapt their metabolism to survive in the absence of AC, although in a way that does not allow them to cope with the stress of nutrient deprivation.

## 1. Introduction

Cutaneous melanoma is one of the most aggressive forms of skin cancer and a major cause of mortality due to its metastatic potential [1]. When it is localised superficially, melanoma undergoes remission in 90% of the cases after surgery; however, advanced melanoma is often resistant to chemotherapy and radiotherapy due to its high genetic and metabolic plasticity, as well as its ability to metabolise chemotherapeutic drugs [2]. The rapidly rising incidence of melanoma over the last decade [3] has prompted an increase in research into the mechanisms involved in its pathogenesis, treatment response and resistance.

Studies have convincingly shown that sphingolipid-mediated signalling affects melanoma evolution by regulating the fate of melanoma cells [4]. Indeed, melanoma exploits aberrant sphingolipid signalling to promote its progression and metastasis. One of the well-known mechanisms used by melanoma to alter sphingolipid function is via over-expression of the lysosomal hydrolase acid ceramidase (*N*-acylsphingosine deacylase), which catalyses the hydrolysis of the pro-apoptotic long-chain ceramides (C14:0, C16:0 and C18:0) into sphingosine and fatty acid [5]. In humans, AC is encoded by the gene *ASAH1*, which maps to chromosome 8 (p21.3–22). Hypofunctional *ASAH1* mutations cause Farber’s disease, a rare lysosomal disorder characterized by articular deformation, subcutaneous nodules and progressive hoarseness [6]. Synthesised in cells as an inactive precursor, AC undergoes self-catalysed proteolysis to yield two subunits, α (13 kDA) and β (40 kDA), which are localised in lysosomes [7]. One of the two products of the AC-catalysed hydrolysis of ceramide, sphingosine, can be converted into sphingosine-1-phosphate (S1P) by sphingosine kinases. Ceramide and S1P regulate the cellular fate in opposite directions. Long-chain ceramides can cause cell cycle arrest, apoptosis and senescence, whereas S1P is anti-apoptotic and favours cell growth, cell motility and angiogenesis [8,9,10]. AC over-expression has been documented in many cancers exhibiting or prone to develop drug-resistance—including melanoma, prostate cancer and glioblastoma—and several AC inhibitors have been developed with the aim of reducing drug resistance [11,12,13]. Recent data show that AC inhibitors or complete ablation of its encoding gene *ASAH1* re-sensitise tumour cells to chemotherapy response [14,15,16]. Despite this evidence, the mechanism of cancer re-sensitisation promoted by AC inhibition is still poorly characterized.

The work of Turner and collaborators pointed to a possible connection between AC and autophagy flux [17]. Autophagy is a homeostatic process that is induced by nutrient deprivation, hypoxia, or other stress factors, and is responsible for the intracellular degradation of damaged cytoplasmic constituents. It involves the formation of a double-membrane structure (the phagophore) where cytoplasmic material is captured to form an autophagosome, which is then targeted to and degraded by lysosomes. Given the important role of autophagy in protein and organelle turnover, it is not surprising that deregulation in this process plays a critical role in metabolic and neurodegenerative disorders, as well as in invasive metastatic melanoma [18] and other forms of cancer [19]. In previous work, we generated a human melanoma cell line in which the gene *ASAH-1* was deleted using CRISPR-Cas9 editing. These AC-null cells exhibited strong down-regulation of Microphthalmia-associated transcription factor (MITF), a key regulator of melanocyte development that plays a central role in melanoma progression [16]. Interestingly, MITF was recently associated with the regulation of lysosomal biogenesis and autophagy through a transcriptional up-regulation of autophagy-related genes [20]. Evidence indicates that lysosomal and autophagic activity are crucial for melanoma progression [21]. While it is well established that genetic mutations that impact AC are the molecular basis for Farber’s disease [6], it is not yet clear how reduced AC activity and lysosomal ceramide accumulation might interfere with autophagy.

In this report, we examined the role of AC in maintaining cellular homeostasis through the regulation of autophagy and mitochondrial biogenesis in the melanoma cell line A375. We show that genetic ablation of AC leads to impaired autophagy and reduces mitochondrial function, conferring several disadvantages to the cells; these results may be leveraged to develop AC inhibitors as adjuvant therapeutics to chemo- and radiotherapy.

## 2. Results

### 2.1. Autophagy Is Active in A375 Melanoma Cells Expressing High Levels of AC

A375 cells were chosen as model of invasive melanoma. As shown in Figure 1a, AC mRNA level is lower in invasive melanomas (A375, HT114, RPMI7951, MM1277) than proliferative melanomas (G361, M14, SK-MEL28). Based on this finding, we decided to focus our study on the highly invasive A375 melanoma cell line, in which AC transcription is comparable to the other invasive melanomas and was easy to select by gene-editing for AC deletions. The full description of A375 AC-null cell line is available in a previous work [16].

To assess whether AC might play a role in the cellular response to nutrient deprivation, which is known to trigger autophagy [22,23,24], we first examined the effects of nutrient deprivation on AC transcription in A375 melanoma cells. Cells were exposed to glucose-free medium for 24 h, and ASAH1 mRNA levels were measured by q RT-PCR. The results show that glucose deprivation caused a significant increase in AC transcription (Figure 1b), this was also confirmed by immunocytochemistry assays (Figure 1c,d). Interestingly, glucose starvation did not affect A375 cell viability (Figure 1e).

Autophagy is a major pathway involved in survival during starvation and plays a key role in chemotherapy resistance [25]. To evaluate the possible contribution of AC to autophagy, we measured autophagy-related protein levels in A375 AC-null cells, which harbour two frameshift mutations in both AC alleles [16]. Under baseline conditions, wild-type A375 cells had 3-times higher levels of the autophagy marker LC3 II compared to A375 AC-null cells (Figure 2a). This difference was further magnified after cell starvation (Figure 2a,d). Interestingly, we did not observe differences in P62 and Beclin-1 expression, indicating that probably autophagy does initiate also in A375 AC-null cells (Figure 2a–c). To verify if the expression of LC3-II correlated to AC levels, we performed a recovery assay by transfecting A375 AC-null cells with a plasmid encoding for AC. Figure 2e shows that AC transfection restores LC3-II levels in A375 AC-null cells. The results suggest that AC expression is important for both baseline and starvation-induced LC3 I-II activity in A375 melanoma cells.

Then, we analyzed autophagic flux by transfecting A375 and A375 AC-null cells with pLC3-EGFP-mRFP. The analysis was performed by counting the number of LC3-RFP puncta (autolysosomes) in basal and starving condition. To better evidence differences in terms of autophagosome accumulation, we also added chloroquine (CHQ) 1 h prior fixation, a potent inhibitor of autolysosome formation. The screening revealed that A375 AC-null cells have less LC3-RFP puncta compared to A375; interestingly, we observed no variation of LC3 puncta when CHQ is added to AC-null cells (Figure 2f,g). These results, together with those illustrated in Figure 2a, indicate that AC deletion decreases autolysosomes. To extend this finding to melanoma cells that accumulate AC (Figure 1a), we transfected AC-siRNA into M14 and SK-MEL-28, melanoma cell lines known to express high levels of AC [15]. As shown in Appendix A, siRNA transfection decreased but did not suppress the levels of AC transcripts. Western blot analysis shown in Appendix A revealed no differences in LC3 II content in both M14 and SK-MEL-28 cells transfected with AC-siRNA. These findings indicate that AC expression, even at low levels, might be sufficient to maintain active autophagy.

### 2.2. A375 AC-Null Cells Undergo Apoptosis after Short-Term Total Nutrient Deprivation

Autophagy plays an important role in cell homeostasis and protects cells by preventing them from entering apoptosis under stress conditions. Because autolysosome formation is defective in A375 AC-null cells (Figure 2), we tested whether these cells might also experience increased cell death following total nutrient starvation. We starved A375 and A375 AC-null cells, transfected or not with pAC, using HBSS medium for 2 h. HBSS did not contain glucose, sera, amino acids, vitamins and other micronutrients. After total starvation, cells were then incubated with a fluorogenic substrate that allows detection of activated caspase 3 and 7 in apoptotic cells, in combination with SYTOX dead-cell stain. As illustrated in Figure 3a, we discriminated living cells (Caspase 3–7-negative/PI-negative), dead cells (Caspase 3–7-negative/PI-positive), apoptotic (Caspase 3–7-positive/PI-negative), and necrotic (Caspase 3–7-positive/PI-positive). As shown in Figure 3b, we observed that 28.2% of A375 AC-null cells stained positive for Caspase 3-/7, compared to 6.9% of control cells. After pAC transfection, we observed that the A375 AC-null Caspase 3/7-positive cell population decreased from 28.2% to 7.5%, thereby getting closer to the values found for A375 (6.9% Caspase 3/7-positive cells). Necrotic cells populations (Q2) were minimally affected by AC levels in both cell lines analysed.

### 2.3. AC Ablation Reduces Mitochondrial Biogenesis by Downregulation of MITF and PGC1α

Autophagy removes dysfunctional mitochondria and increases the number of apoptotic events after nutrient deprivation. Prompted by the evidence that ablation of AC impairs autophagy, we hypothesised a possible mitochondrial dysfunction in A375 AC-null cells that might explain the increased apoptotic induction. To probe this idea, we first measured the expression of PGC1α, a transcription co-activator that stimulates mitochondrial biogenesis. We evaluated the expression of PGC1α also because its transcription in melanoma cells is activated by MITF, which is strongly downregulated in A375 AC-null cells [16]. We measured MITF transcription by q RT-PCR in A375 AC-null and WT cells. As shown in Figure 4a, MITF mRNA levels were strongly downregulated in A375 AC-null cells compared to wild-type cells. To better understand if MITF expression depended on AC expression, we transfected A375 AC-null cells with pAC. As shown, AC recovery increases MITF expression by 5 times in A375 AC-null cells. Consistently with such profound MITF downregulation in A375 AC-null cells, PGC1α expression was also decreased (~2-fold reduction), compared to A375 control cells. To further confirm the crosstalk of AC with MITF and PGC1α, we observed a 2-fold increase in PGC1α levels after pAC recovery transfection (Figure 4b).

Mitochondrial dynamics are dictated by a series of fusion and fissions events of the mitochondrial network. This mitochondrial plasticity is fundamental in maintaining the integrity and the functionality of the mitochondria [26]. The induction of mitochondrial fusion allows the formation of extended and interconnected mitochondrial networks, whereas a shift towards fission generates morphologically and functionally distinct organelles. To evaluate differences in mitochondrial dynamics, we first quantified the total mitochondria mass in A375 and A375 AC-null cells. We performed flow cytometry analysis on A375 and A375 AC-null cells stained by MitoTracker Green, a fluorescent marker that labels all mitochondria (Figure 4c). The analysis showed that the average number of mitochondria did not change significantly between A375 and A375 AC-null cells (Figure 4d).

Then, we probed expression levels of MFN-1 and DRP-1 genes. These genes, among others, regulate mitochondrial fusion (MFN-1) and fission (DRP-1), respectively. As shown in Figure 4e,f, we found that MFN-1 mRNA expression up-regulated in A375 AC-null cells, as we observed an increase in DRP-1 expression, this was not statistically relevant. Collectively, qRT-PCRs indicate a derangement of mitochondrial network dynamics in A375 AC-null cells. Starting from this finding, we probed AC-null cells for mitochondria integrity and membrane polarization, using high content confocal microscopy screening. Briefly, cells were stained with MitoTracker Red CMXros, which labels mitochondria based on their ROS production. This assay is used to measure increased mitochondrial ROS, which overwhelms normal antioxidant defences and set up the opening of the mitochondrial permeability transition pore, which leads to further membrane depolarization and release of pro-apoptotic proteins into the cytosol. We observed that A375 AC-null cells exhibited a significant increase in mitochondrial ROS intensity compared to A375 control cells (Figure 5a,b). Then, mitochondria texture analysis was performed within the Harmony Software using the “SER Features” method. SER analysis was performed at 2px resolution to evaluate the alteration in mitochondria texture and their overall morphology. Briefly, CMXros-stained mitochondria were segmented using the SER algorithm, as shown in Figure 5c, using the following parameters: hole, edge, ridge, valley, saddle, roundness and length. SER texture-based analysis read out revealed that A375 AC-null cells have profound alterations of mitochondria structure and shape in A375 AC-null cells compared to their WT counterparts (Figure 5d–h), while the roundness and length of mitochondria remained unchanged (Figure 5i,j).

## 3. Discussion

It is well established that AC inhibitors reduce resistance to chemotherapy [5] and radiotherapy of melanoma and several other types of cancer [14,16]. For this reason, in recent years, different compounds were designed, with the aim to achieve a novel kind of adjuvant therapy for aggressive types of cancer that do not respond to conventional drugs. Although several research groups brilliantly demonstrated that AC inhibition, whether by drugs, siRNA or gene-editing techniques, leads to an overall sensitization to therapy, it is still unclear which mechanism accounts for this evidence [27,28].

The present work sheds light on a mechanism whereby AC expression hinders metabolic equilibrium of melanoma cells through autophagy impairment and mitochondrial dysfunction. Two observations prompted us to study the relationship between autophagy and AC expression: (1) the first evidence was provided by Turner and colleagues [17], who observed that autophagy enhanced resistance of prostate cancer cells to endogenous ceramides, which normally lead cells to death. In the same year, Bedia and colleagues showed that Dacarbazine, a common drug used to treat metastatic melanoma, reduced AC activity in A375 melanoma cells [5]. Surprisingly, the latter effect was paralleled by a decrease in autophagic activity. (2) The second observation is provided by genetics: mutations of the ASAH1 gene, which encodes for AC, cause a lysosomal storage disease (LSD) known as Farber’s syndrome. Interestingly, patients who suffer from LSDs exhibit dysfunctional mitochondria, impairment of autophagy and accumulation of cytoplasmic protein aggregates [29,30]. Moreover, recently Coazzoli et al. showed that another key enzyme in sphingolipid metabolism, acid sphingomyelinase, has an important role in the regulation of mitochondrial function and morphology in melanoma cells [31].

In LSDs, a significant number of autophagosomes remains in the cytosol, suggesting incomplete autophagy [32]. Therefore, autophagy impairment observed in LSDs might depend on the last phase of autophagy, in which autophagosomes are targeted by lysosomes to form autolysosomes [25]. Consistently with these observations, our data show that autophagy is impaired also in A375 AC-null cells, where LC3 isoform II is decreased, but is restored after the transfection of AC in trans. However, one of the major autophagy-initiator proteins, Beclin-1, is expressed at the same levels in A375 AC-null cells and A375 control cells, suggesting that autophagy is activated upstream in A375 AC-null cells but fails to complete the cargo digestion. Indeed, we observed that A375 AC-null cells have impaired LC3-RFP puncta, confirming a dysfunction in later phases of autophagy in these mutant cells. Unfortunately, siRNA transfection of AC-overexpressing SK-MEL-28 and M14 did not cause autophagic dysfunction observed by genetic ablation. The reason behind this unexpected result might be incomplete AC elimination: despite an 80% reduction in AC mRNA levels obtained by siRNA transfection of M14 and SK-MEL-28, what remains might be enough to avoid long-chain ceramide accumulation in lysosomes.

Recent evidence points to a relationship between MITF expression and autophagy induction in melanoma cells [20]. Autophagy and apoptosis are finely interconnected [33], and the inability to activate autophagy during a given stress condition activates apoptosis. Our previous work [16] had uncovered considerable MITF downregulation in AC-null cells. The results provided in this study show that MITF downregulation is still found in A375 AC-null cells even after years of propagation. This indicates that downregulation of MITF caused by AC depletion cannot be compensated in A375 cells. In the present study, AC ablation caused a 3-fold decrease in MITF expression by A375 AC-null cells. Consistently, we also found that, after short-term nutrient deprivation, ~30% of A375 AC-null cells undergo apoptosis compared to ~6% of A375 cells. This remarkable difference led us to think that A375 AC-null cells might have a fragile metabolic equilibrium that hinders their ability to mount an adaptive response. To test this hypothesis, bearing in mind that Farber’s disease patients accumulate C16 and C18 ceramides inside mitochondria, we explored mitochondria biogenesis, structure, and function in A375 AC-null cells. We first counted the total number of mitochondria in A375 AC-null and A375 cells by flow cytometry. This analysis showed no variation in the numbers of mitochondria between the two cell populations, as expected after finding that autophagy is hampered. Nonetheless, analysis of mitochondrial shape and function revealed that AC-null cells exhibit statistically significant mitochondrial membrane ROS accumulation. Mitochondria also appear to be different in shape and organisation in AC-null cells, compared to their wild-type counterparts. Although further studies are necessary to speculate on these observations, these features are reminiscent of the mitochondrial alterations documented in persons with Farber’s disease [32].

In melanoma, MITF strictly controls the levels of the main regulator of mitochondria biogenesis PGC1α. A study by Vazquez et al. [34] showed that melanoma cell lines with high levels of PGC1α have more mitochondria. Consistent with this finding, we show a strong downregulation of PGC1α in A375 AC-null cells, which confirms that these cells are unable to cope with nutrient deprivation by generating new mitochondria. The combination of impairment in mitochondrial biogenesis with the reduction in autophagy, which leads to accumulation of dysfunctional mitochondria in AC-null cells, might explain why AC-depleted A375 cells cannot survive under nutrient deprivation, even for a short period of time.

In conclusion, the present results show that, (1) AC-null melanoma cells have reduced autolysosomes, possibly due to local accumulation of non-metabolised ceramides. This defect in autophagic flux may result in the inability to dismantle cellular organelles and molecules to produce energy. Moreover, impaired autophagy implies impaired mitophagy, which might explain why damaged mitochondria accumulate in AC-null cells. (2) AC deletion causes downregulation of MITF and PGC1α. This may alter both the plasticity of melanoma cells under stress (via MITF) and their ability to produce new and functional mitochondria (via PGC1α). Together, these findings suggest that AC-null melanoma cells change and adapt their metabolism to survive in the absence of AC, although in a way that does not allow them to cope with the stress of nutrient deprivation.

## 4. Materials and Methods

### 4.1. Cell Culture, Treatments and Transfection

Human melanoma A375, M14 and SK-Mel-28 cells were purchased from American Type Culture Collection (Manassas, VA 20110, USA), and cultured at 37 °C in 5% CO2 in Dulbecco-Modified Eagle’s Medium (DMEM) supplemented with 10% fetal bovine serum (FBS), 100 U/mL penicillin and 100 µg/mL streptomycin. A375 knock-out for *ASAH-1* (A375-AC null) cells were obtained by CRISPR/Cas9 editing, as described before [16]. The recovery of AC in A375 AC-null cells was performed using a commercial plasmid containing the ASAH1 cDNA under control of the CMV promoter (RG212434 Origene, Rockville, USA, MD 20850). Cells were transfected using Lipofectamine LTX and PLUS reagents (15338100, Thermo Fisher Scientific, Waltham, MA 02451, USA) according to the manufacturer’s instructions. AC silencing was performed by transfecting cells with AC-siRNA (ASO2GM43, Thermo Fisher Scientific, Waltham, MA 02451, USA) using Lipofectamine RNAiMAX according to the manufacturer’s instructions (13778075, Invitrogen, Waltham, MA 02451, USA).

### 4.2. Western Blotting

Melanoma cell lines (3.5 × 10^5^ cells/well) were plated in 6-well dishes and, the following day, the regular medium was replaced with either “starving medium” (DMEM without glucose and FBS) or complete medium. Then 24 h later, the cells were lysed in RIPA buffer (R0278, Sigma-Aldrich, St. Louis, MO 63103, USA) supplemented with protease inhibitors (Halt^TM^ Protease Inhibitor Cocktail 100×, 87785, Thermo Fisher Scientific, MA, UK). Cell suspensions were centrifuged at 13.000× *g* at 4 °C for 20 min. Protein concentration was measured using the Bradford method (B6916, Sigma-Aldrich, St. Louis, MO 63103, USA). Equal amounts of protein (30 µg) were mixed with NuPAGE LDS sample buffer (NP0007, Thermo Fisher Scientific, Waltham, MA 02451, USA) and separated on 4–12% NuPAGE Bis-Tris gels (NP0321BOX, Thermo Fisher Scientific, Waltham, MA 02451, USA) and then transferred to Amesham Protan Nitrocellulose Membranes 0.45 µm (GE10600002, GE Healthcare, Marlborough, MA, USA). The membranes were washed with phosphate buffer saline (PBS) containing 0.1% Tween 20 (P9416, Sigma-Aldrich, St. Louis, MO 63103, USA) (Tween 0.1%-PBS) and blocked with 5% BSA in Tween 0.1%-PBS at room temperature (RT) for 1 h. Then the membrane was incubated with primary antibodies at 4 °C overnight for anti-Beclin-1 (1:1000, BK 3738S, Cell Signaling, Danvers, MA 01923, USA), anti-P62 (1:1000, ab56416, Abcam, Cambridge, UK), anti-LC3 I-II (1:1000, L7543, Sigma-Aldrich, St. Louis, MO 63103, USA). As a loading control, we used rabbit anti-β-Actin (1:1000, A2066 Sigma-Aldrich, St. Louis, MO 63103, USA).

### 4.3. Cell Viability

Cells were plated (2 × 10^4^ cells/well) in 96-well plates and incubated overnight. The cells were then incubated in starving medium for 24 h and their viability was assessed using Alamar Blue (BUF012A, Bio-Rad, Hercules, CA, USA) assay according to the manufacturer’s instruction.

### 4.4. RNA Extraction and Quantitative Real Time-PCR (qPCR)

Cells (3.5 × 10^5^ cell/well) were plated and incubated overnight in 6-well plates. The following day, the complete medium was replaced with starving medium for 24 h. Total RNA was extracted with TRI Reagent (T9424, Sigma-Aldrich, St. Louis, MO 63103, USA) according to the manufacturer’s instructions. Samples were treated with DNAseI (DNAseI SuperScript IV Reverse Transcriptase (18090010, Thermo Fisher Scientific, Waltham, MA 0245, USA), according to the manufacturer’s instructions. First-strand cDNA was amplified using SsoAdvanced Universal SYBR Green Supermix (1725271, Bio-Rad, Hercules, CA, USA). *ASAH1* gene: *F: 5′-AGTTGCGTCGCCTTAGTCCT-3′; R: 5′-TGCACCTCTGTACGTTGGTC-3′; PPARGC1A gene: F: 5′-GAGTCTGTATGGAGTGACATCG-3′; R: 5′-TGTCTGATACCAAGTCGTTCAC-3′. MITF: F: 5′-CTCACCATCAGCAACTCCTG-3′; R: 5′-GATTGTCCTTTTTCTGCCTCTC-3′. MFN1: F: 5′-CCTACTGCTCCTTCTAACCCA-3′; R: 5′-AGGGACGCCAATCCTGTGA-3′. DRP1: F: 5′GCTGGATCACGGGACAAGTTAA-3′; R: 5′-TGCCTGTTGTTGGTTCCTGAC-3′.* Results were normalized using β-Actin. Relative quantification of gene expression was calculated by the 2-ΔΔCt method.

### 4.5. Confocal Microscopy and Flow Cytometry

A375 cells were plated (5 × 10^3^ cells) in 96-well Cell-carrier Ultra. The following day, cells were incubated in normal or starving condition for 24 h. After then, cells were fixed and stained for AC and visualised using Operetta High Content Imaging using 40× objective. The intensity of AC in the cytoplasm was analysed using Harmony High Content Imaging and Analysis Software. A375 and A375 AC-null cells were plated (2 × 10^4^ cells) in 96-well plates and stained with Mitotracker CMX ROS (50 nM, MitoTracker^TM^ Red CMXRos, M7512, Thermo Fisher Scientific, Waltham, MA 02451, USA) and incubated for 30 min at 37 °C. After staining, the medium was replaced with fresh medium and the cells were visualised using Operetta High Content Imaging System. Cells were acquired using 40× objective.

To assess autophagy flux, we seeded A375 and A375 AC-null in 96-well Cell Carrier ultra, and the day after, we transfected the cells with ptfLC3-EGFP-mRFP (Addgene plasmid #21074) using Lipofectamine LTX and Plus reagents. After 24 h of transfection, cells were starved with starving medium for 6 h (+/− chloroquine 1 µM for 1 h). Then, the cells were washed in PBS and fixed with PFA 4%. Cells were analysed by Operetta High Con-tent Imaging System, acquired using 63× objective. For Flow Cytometry, cells (5 × 10^4^ cells/well) were seeded in 24-well plates and incubated overnight. The following day, samples were stained with 50 nM Mitotracker Green for 30 min following the manufacturer’s instruction. After staining, the cells were suspended by trypsin incubation, washed with PBS, and analysed by flow cytometry using an ATTUNE NXT Flow Cytometer (Thermo Fisher Scientific, Waltham, MA, USA).

### 4.6. CellEvent Caspase-3/7 Assay

A375 and A375 AC-null cells were plated (5 × 10^4^ cells/well) in 12-well plates. Cells were transfected or not with pAC plasmid and after 24 h were incubated with Hank’s balanced salt solution (HBSS) (14025, Sigma-Aldrich, St. Louis, MO 63103, USA) for 2 h. Apoptosis events were detected using CellEvent^TM^ Caspase-3/7 Green Flow Cytometry Assay Kit (C10427, Thermo Fisher Scientific, Waltham, MA 02451, USA) according to the manufacturer’s instructions. Briefly, 1 µL of CellEvent Caspase-3/7 Green Detection Reagent was added to the samples and incubated for 25 min at 37 °C. One µL of SYTOX AADvanced was then added to each sample and incubated for 5 min at 37 °C. Samples were analysed using an ATTUNE NXT Flow Cytometer.

### 4.7. Statistical Analysis

Statistical significance of differences between the groups was performed using Student’s *t-*test or one-way/two-way ANOVA, followed by Bonferroni post-test (multiple comparisons). Differences between groups were considered statistically significant at *p* values < 0.05. Results are expressed as mean ± SD of two or three independent experiments performed in three technical replicates each.

### 4.8. High-Content Imaging of Mitochondria and Autophagosomes

Imaging experiments were performed using an Operetta CLS high-content imaging device (PerkinElmer, Hamburg, Germany), and analysed with Harmony 4.6 software (PerkinElmer). To investigate LC3-II red spots, we analysed cells using 63x magnification, taking 25 fields per sample in biological and technical triplicates. Data were analysed using the following building blocks: 1- Find Nuclei, 2- Find Cytoplasm of transfected cells (RFP- and GFP-positive) 3- Find RFP+ spots. Mitochondria morphology was first assessed analysing the mitochondria intensity of MitoSOX using the following building blocks: 1—Find Nuclei, 2—Find Cytoplasm (MitoSOX+) 3—Find MitoSOX+ spots (mitochondria) 4—Calculate Intensity Properties on “mitochondria” population. Then, SER analysis was performed to evaluate the alteration in mitochondria morphology. Briefly, cells were segmented using the SER algorithm for the following parameters: hole, edge, ridge, valley, saddle, roundness and length. SER analysis allows texture-based read out of pixel distributions using fluorescence based or brightfield images. For all types of quantitative image analysis, 27 fields of views (FoV) were acquired for each replicates per condition.

## Figures and Tables

**Figure 1 ijms-22-03247-f001:**
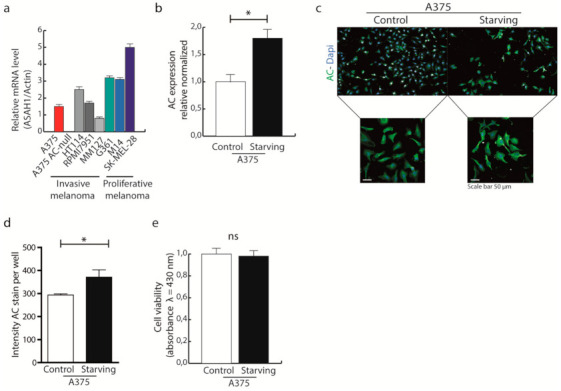
AC expression increases during starvation. (**a**) AC expression in invasive melanomas (A375, HT114, RPMI7951, MM1277) and proliferative melanomas (G361, M14, SK-MEL28). (**b**) A375 melanoma cells were incubated with starving medium for 24 h, AC transcription was measured by Real-time PCR. Results are expressed as mean ± SD, with each experiment performed with two biological and three technical replicates (* *p* < 0.05, Student’s *t-*test). (**c**) A375 were incubated in normal or starving conditions for 24 h. Cells were then fixed, stained for AC and visualised using Operetta High Content Imaging System. Staining intensity was analysed using Harmony Software. (**d**) Analysis of AC intensity in A375 cells exposed or not to starving conditions. Results are expressed as mean ± SD, with each experiment performed with two biological and three technical replicates (* *p* < 0.05, Student’s *t-*test). (**e**) A375 cells were incubated with starving medium for 24 h and cell viability was assessed using the Alamar Blue assay (λ = 430 nm). Results are expressed as mean ± SD, with each experiment performed with two biological and three technical replicates (*p* < 0.05, Student’s *t-*test).

**Figure 2 ijms-22-03247-f002:**
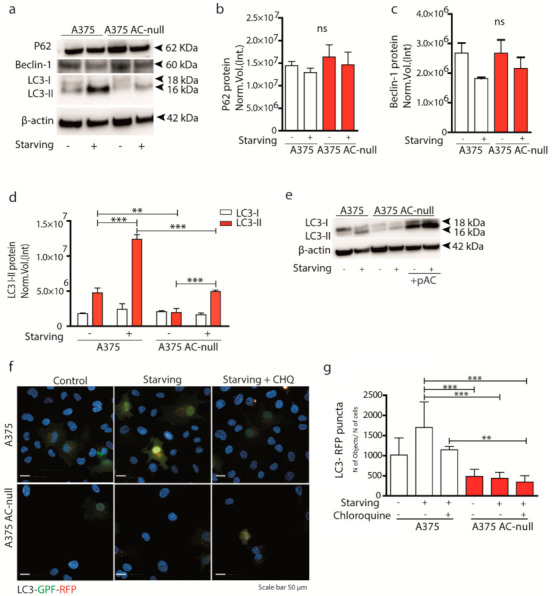
AC ablation deregulates autophagy during glucose starvation. (**a**) Western blotting with anti-P62, anti-Beclin-1 and anti-LC3 I-II antibodies on A375 and A375 AC-null cell lysates under baseline conditions and after 24 h of glucose starvation. (**b**–**d**) P62, Beclin-1 and LC3 I-II Western blot quantification on A375 and A375 AC-null cells, data are expressed as mean ± SD (** *p* < 0.01; *** *p* < 0.001, Two Way-ANOVA, with Bonferroni post hoc test). (**e**) Western blot showing a recovery assay by transfecting A375 AC-null cells with pAC plasmid. Transfection restored the amount of LC3-II to basal A375 levels. (**f**) A375 and A375 AC-null cells were transfected with ptfLC3-GFP-mRFP. After 24 h of transfection, cells were starved for 6 h and treated or not with chloroquine 1 µM for 1 h. Autolysosome (RFP) and autophagosomes (GFP + RFP) were visualised using Operetta High Content Imaging System at 63× magnification. LC3 puncta were counted by Harmony software. Scale bar: 50 µm. (**g**) Quantification of LC3-RFP puncta (autolysosomes) using the following formula: number of LC3 puncta/number of transfected cells per field. Results are expressed as mean ± SD. Experiments are performed in three biological replicates (** *p* < 0.001, *** *p* < 0.0001; One Way ANOVA, Bonferroni post hoc) around 5.10^4^ cells were analysed per experimental groups.

**Figure 3 ijms-22-03247-f003:**
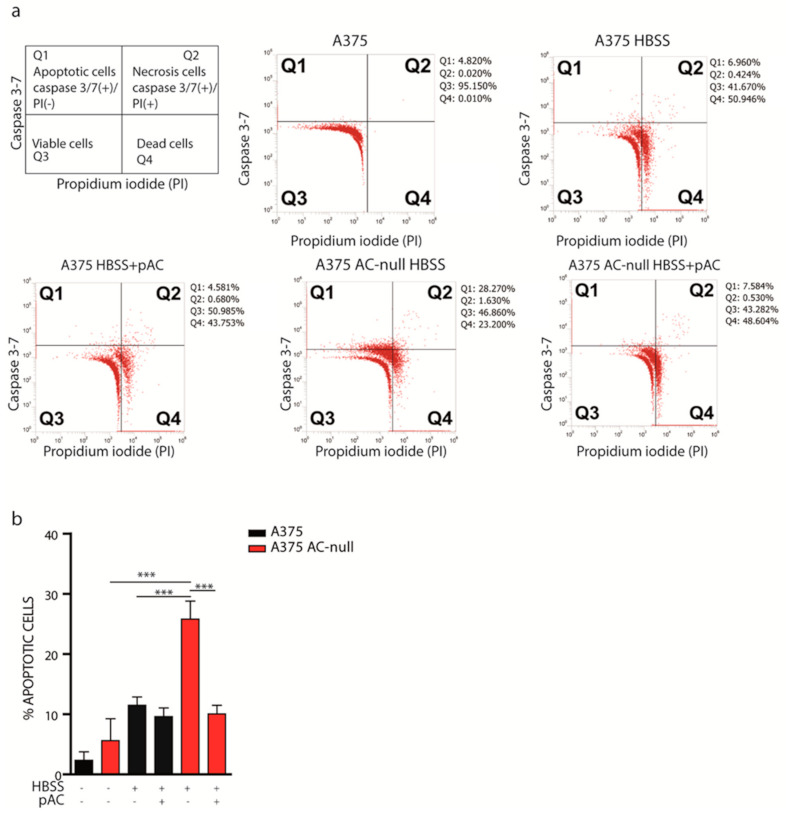
AC ablation promotes apoptosis after total nutrient deprivation. (**a**) A375 and A375 AC-null cells were transfected or not with pAC and after 24 h were starved with HBSS medium for 2 h. Cells were then harvested, stained with Caspase 3-7/SYTOX (propidium iodide), and analyzed by flow cytometry as presented in the illustration. (**b**) Flow cytometry analysis performed on cells in Q1 show an increased number of apoptotic cells in A375 AC-null cells. Moreover, recovery transfection using pAC decreased the amount of apoptotic cells in A375 AC-null cells, while no substantial differences were observed in WT A375 cells, transfected or not with pAC. Experiments were performed in three biological and technical replicates, data are expressed as mean ± SD (*** *p* < 0.001, Student’s *t-*test).

**Figure 4 ijms-22-03247-f004:**
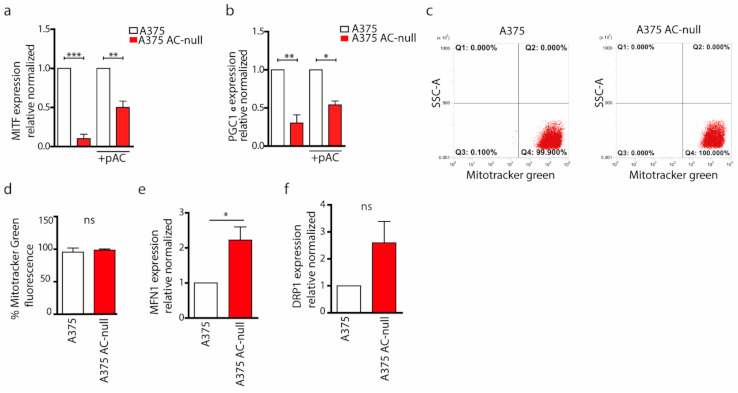
AC ablation results in altered MITF/PGC1α signaling. (**a**) Cells were transfected or not with pAC. Real-time PCR of MITF on A375 and A375 AC-null cells were performed. Results are expressed as mean ± SD, with each experiment performed three biological and technical triplicates (** *p* < 0.01, *** *p* < 0.001, One Way ANOVA and Bonferroni post hoc test). (**b**) Real-time PCR of PGC1α on A375 and A375 AC-null cells, transfected or not with pAC. Results are expressed as mean ± SD, with each experiment performed three biological and technical triplicates (** *p* < 0.01, *** *p* < 0.001, One Way ANOVA and Bonferroni post hoc test). (**c**) Mitochondria in A375 and A375 AC-null cells were labelled with MitoTracker Green for 30′ at 37 °C. Cells were then washed and analysed by flow cytometry. (**d**) Statistical analysis of MitoTracker Green fluorescence intensity on A375 AC-null cells compared to A375 WT cells. Data are expressed as mean ± SD and experiments were performed in three biological and technical replicates (Student’s *t*-test). (**e**,**f**) Real-time PCR of MFN-1 and DRP-1 on A375 and A375 AC-null. Results are expressed as mean ± SD, with each experiment performed three biological and technical triplicates (* *p* < 0.05 Student’s *t-*test).

**Figure 5 ijms-22-03247-f005:**
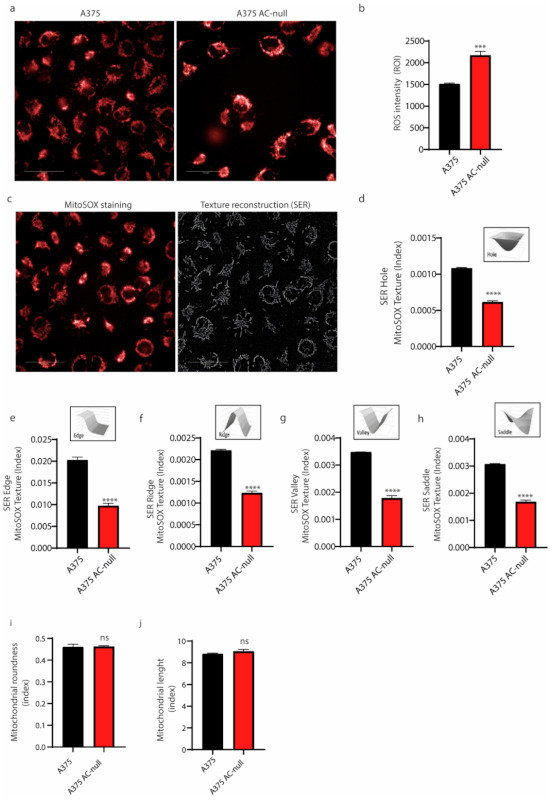
Mitochondria organization in A375 AC-null cells. (**a**) Mitochondrial membrane potential staining with MitoSOX dye was analyzed by high-content confocal screening using 40× water objectives, scale bar: 50 µm. (**b**) MitoSOX intensity analysis on both cell lines revealed increased ROS accumulation in A375 AC-null cells. Seventy-two fields were analyzed in three technical and biological triplicates. Data are expressed as mean ± SD, statistical analysis was performed using Student’s *t-*test, *** *p* < 0.0001. (**c**) Representative image of SER segmentation of MitoSOX mitochondria staining. (**d**–**j**) SER segmentation analyses performed using Student’s *t-*test on morphological properties of mitochondria in A375 and A375 AC-null cells. The analysis revealed profound alterations of mitochondria structure and shape in A375 AC-null cells compared to their WT counterparts (**d**–**h**). The roundness and length of single mitochondria remained unchanged (**i**,**j**). **** *p* < 0.00001; ns = not statistically significant.

## Data Availability

The data presented in this study are available upon request to corresponding author, due to the large dimension of the high-content dataset.

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
