# Peer review of "Ablation of Acid Ceramidase Impairs Autophagy and Mitochondria Activity in Melanoma Cells"

_ijms, 2021, doi:10.3390/ijms22063247_

Round 1

Reviewer 1 Report

Lai et al. present a study demonstrating that the ablation of acid ceramidase causes impaired autophagy and mitochondria dysfunction in A375 cells. Overall, the study showed good data quality, and the manuscript is prepared in a proper format of IJMS. However, most of the study was performed in one cell lines, and some data could not support their conclusions. The comments for the authors are listed as follows:

  1. The authors did not explain why they choose A375 cell in this study. The comparison of the mRNA and protein levels of AC in different melanoma cell lines are necessary in this study.
  2. In Supplementary Figure 1a, how come there are no error bars, and the results were significant by three biological repeats?
  3. In Supplementary Figure 1b, although siRNA did not fully suppress the levels of AC transcripts, how about the protein levels of AC as well as the levels of ceramide, sphingosine or S1P ? How come AC knock-down increased LC3II levels shown in Supplementary 1c? This is not consistent to the results of A375 cells.
  4. The title of Supplementary Figure 1 was wrong.
  5. In Figure 1a and 1c, it would be nice to show the protein levels of AC.
  6. In Figure 1g, the band width of +AC were not equivalent to other groups. Is it from the same SDS-PAGE?
  7. The experimental design of Figure 2 was not proper, and what is the definition of Q4? It would be nice to use Annexin V/PI staining to distinguish early apoptosis, late apoptosis and necrosis. Meanwhile, it would be nice to include control groups, where the cells were not starved by HBSS.
  8. In line 143, the authors claim that reduced mitochondrial biogenesis was resulted from downregulation of MITF and PGC1alpha. But the authors did not overexpress MITF and PGC1alpha to see if there will be reversal effects on mitochondrial dysfunction.
  9. In line 171, the authors mentioned the probed expression levels of MNF-1, -2, OPA-1 and DRP-1 genes. But the results only showed MFN-1 and DRP-1 gene expression.

Author Response

Manuscript ID: ijms-1094810

Point by point replies to reviewer 1

We thank the reviewer for its detailed work that substantially improved our manuscript

Lai et al. present a study demonstrating that the ablation of acid ceramidase causes impaired autophagy and mitochondria dysfunction in A375 cells. Overall, the study showed good data quality, and the manuscript is prepared in a proper format of IJMS. However, most of the study was performed in one cell lines, and some data could not support their conclusions. The comments for the authors are listed as follows:

  1. The authors did not explain why they choose A375 cell in this study. The comparison of the mRNA and protein levels of AC in different melanoma cell lines are necessary in this study.

Done. Lines 94-97 “we decided to focus our study on the highly invasive A375 melanoma cell line, in which AC transcription is comparable to the other invasive melanomas and were easy to select by gene-editing for AC deletions. The full description of A375 AC-null cell line, is available in a previous work [16].”

  1. In Supplementary Figure 1a, how come there are no error bars, and the results were significant by three biological repeats?

We are sorry for the refuse, the correct Supplementary figure 1 is now added.

  1. In Supplementary Figure 1b, although siRNA did not fully suppress the levels of AC transcripts, how about the protein levels of AC as well as the levels of ceramide, sphingosine or S1P ? How come AC knock-down increased LC3II levels shown in Supplementary 1c? This is not consistent to the results of A375 cells.

To avoid republishing of existing data we added details about the levels of sphingosine, S1P and of ceramides in A375 and A375 AC-null  provided in a previous work:  lines 316-317 “Lipidomic profiles of A375 and A375 AC-null cells are available in a previous work published by our team [16]”

  1. The title of Supplementary Figure 1 was wrong.

Correct line 310

  1. In Figure 1a and 1c, it would be nice to show the protein levels of AC.

Protein levels of AC in A375 and A375 AC-null cells are described in our previous work, to better clarify the AC expression before and after starving we added an high-content quantification of AC in A375 cells (Figure 1c). lines 102-104 was added “caused a significant increase in AC transcription (Figure 1b), this was also confirmed by immunocytochemistry analysis (figure 1c-d). glucose starvation did not affected cell viability (Figure 1e).”

  1. In Figure 1g, the band width of +AC were not equivalent to other groups. Is it from the same SDS-PAGE?

Figure 1g is now moved to figure 2e, that shows in the same SDS-PAGE the recovery of LC3-II

  1. The experimental design of Figure 2 was not proper, and what is the definition of Q4? It would be nice to use Annexin V/PI staining to distinguish early apoptosis, late apoptosis and necrosis. Meanwhile, it would be nice to include control groups, where the cells were not starved by HBSS.

We changed the dot-plot gate to clarify the analysis, definition of Q4 is added. Lines 145-147 were added “We starved A375 and A375 AC-null transfected or not with pAC, using HBSS medium for 2 h. HBSS did not contain  glucose, sera, amino acids, vitamins and other micronutrients.” we discriminated living cells (Caspase 3-7 negative /PI negative), dead cells (Caspase 3-7 negative /PI positive), apoptotic (Caspase 3-7positive /PI negative), and necrotic (Caspase 3-7 positive/PI positive).”

  1. In line 143, the authors claim that reduced mitochondrial biogenesis was resulted from downregulation of MITF and PGC1alpha. But the authors did not overexpress MITF and PGC1alpha to see if there will be reversal effects on mitochondrial dysfunction.

We agree with reviewer, we plan to dedicate an entire work on mitochondrial dysfunction after AC ablation and the suggested experiments that will need several molecular clonings and validations, will be part of it.

  1. In line 171, the authors mentioned the probed expression levels of MNF-1, -2, OPA-1 and DRP-1 genes. But the results only showed MFN-1 and DRP-1 gene expression.

We corrected the text removing OPA-1 and MFN-2 from the text and figure 3 legend.  New lines were added to discuss implication of MFN-1 and OPA-1 expression, lines 183-186 “Then, we probed expression levels of MFN-1 and DRP-1 genes. These genes, among others, regulate mitochondrial fusion (MFN-1) and fission (DRP-1), respectively. As shown in Figure 4e-f, we found MFN-1 mRNA expression up-regulated in A375 AC-null cells, as we observed an increase in DRP-1 expression, this was not statistically relevant”

Reviewer 2 Report

In this paper, the authors build on previous work on the role of ceramidase in melanoma. Ceramidase is an enzyme overexpressed in different cancers which catalyzes the hydrolysis of pro-apoptotic long-chain ceramides into sphingosine and fatty acid. Therefore, ceramidase has an anti-apoptotic effect. The authors suggest ceramidase has a role in autophagy and it can also regulate mitochondrial function. These studies present very conflicting results and additional experiments/controls are required.

Major comments:

  1. The authors show transcriptional changes in AC after nutrient starvation, but they should also show protein levels by WB. Furthermore, given the suggested role of AC in autophagy, immunofluorescence for AC would be recommended to define the cellular location of the protein in these conditions.
  2. The authors show no differences in cell viability in nutrient depleted conditions (Figure 1B). How does this data correlate with the antiapoptotic role of AC? On the other hand, the data in figure 2 shows differences in apoptosis after 2h of starvation in A375 cells. How do the authors reconcile this data with the lack of differences in cell viability at 24h?
  3. There are important caveats in figure 1 regarding the role of AC in autophagy. The decrease in the LC3-II band observed in A375 AC-null cells (Figure 1C) could indicate a decrease in autophagy or an accelerated flux through the pathway that depletes autophagosomes. The best way to assess autophagic flux is to use bafilomycin or chloroquine in basal conditions and in nutrient starved conditions to block autophagosome degradation and assess intensity of LC3-I/LC3-II. The rescue experiment with pAC is interesting but this western blot is very problematic. All samples should be run in the same blot to get a better estimate of LC3-I and LC3-II in comparison to A375 and A375-null cells. Furthermore, this WB seems to duplicate actin from Figure 1C but LC3 seems to be from a different blot? Finally, the authors should indicate the conditions for the immunofluorescence in Figure 1h (basal or starved). We recommend staining for LC3 instead of/in addition to lysotracker to check for LC3 puncta. We recommend doing this experiment in basal and nutrient starved conditions +/- chloroquine or bafilomycin. This will be a better indicator of autophagic flux in nutrient starved conditions.
  4. Figure 2 suggests AC ablation promotes apoptosis after nutrient deprivation. For any cell-death experiment by flow, controls in basal conditions should be provided for all cell lines. The authors focus on the reduction in apoptotic cells with pAC rescue (24.8 to 9%) but the % of viable cells is only 10-20% in all conditions. These results don’t match with the data presented in Figure 1 suggesting that nutrient starvation does not impact cell viability. Controls in basal conditions would help to assess the gating strategy and to differentiate if those PI+ cells are really necrotic cells or artifacts.
  5. In figure 3, the authors show a decrease in MITF and PGC1 expression in AC null cells which would be suggestive of decreased mitochondrial biogenesis. Next, they evaluate mitochondrial mass with mitotracker green and mitochondrial polarization with mitotracker red. The authors quantify mito green by FACS and they show no differences between cell lines. Nevertheless, the IF shows higher staining in the null cells. WB for mitochondrial markers such as Tom20 or others are encouraged. Furthermore, based on the IF results, the authors suggest mitochondrial morphology is altered in AC KO. This is interesting but a quantification of % of fragmented vs elongated mitochondria would be required. Also, how do the authors interpret an increase in MFN1 and DRP1 expression by RT-PCR with the differences in mitochondrial elongation?

Author Response

Manuscript ID: ijms-1094810

Point by point replies to reviewer 2

Comments and Suggestions for Authors

In this paper, the authors build on previous work on the role of ceramidase in melanoma. Ceramidase is an enzyme overexpressed in different cancers which catalyzes the hydrolysis of pro-apoptotic long-chain ceramides into sphingosine and fatty acid. Therefore, ceramidase has an anti-apoptotic effect. The authors suggest ceramidase has a role in autophagy and it can also regulate mitochondrial function. These studies present very conflicting results and additional experiments/controls are required.

We thank the reviewer for its detailed revision that substantially increased the quality of our manuscript

Major comments:

The authors show transcriptional changes in AC after nutrient starvation, but they should also show protein levels by WB. Furthermore, given the suggested role of AC in autophagy, immunofluorescence for AC would be recommended to define the cellular location of the protein in these conditions.

Protein levels of AC in A375 and A375 AC-null cells are described in our previous work, cited [16] in the text. to better clarify the AC expression before and after starving we added an high-content quantification of AC in A375 cells (Figure 1c) that confirms the increased AC expression after starvation observed using qRT-PCR. Lines 101-105 “Cells were exposed to glucose free medium for 24 h, and ASAH1 mRNA levels were measured by q RT-PCR. The results show that glucose deprivation caused a significant increase in AC transcription (Figure 1b), this was also confirmed by immunocytochemistry assays (figure 1c-d). Interestingly, glucose starvation did not affected A375 cell viability (Figure 1e).”

The authors show no differences in cell viability in nutrient depleted conditions (Figure 1B). How does this data correlate with the antiapoptotic role of AC? On the other hand, the data in figure 2 shows differences in apoptosis after 2h of starvation in A375 cells. How do the authors reconcile this data with the lack of differences in cell viability at 24h?

We apologise for the lack of clarity, we tested two different types of starving: the first one, described in figure 1 and 2, was performed using a glucose free DMEM media. The second one was performed using HBSS media (figure 3), without any kind of nutrients (no sera, no glucose, no amino acids, no vitamins). We added in legends and in the results the differences of both starving procedures. Lines 140- 142 “We starved A375 and A375 AC-null cells, transfected or not with pAC, using HBSS medium for 2 h. HBSS did not contain glucose, sera, amino acids, vitamins and other micronutrients.”.

There are important caveats in figure 1 regarding the role of AC in autophagy. The decrease in the LC3-II band observed in A375 AC-null cells (Figure 1C) could indicate a decrease in autophagy or an accelerated flux through the pathway that depletes autophagosomes. The best way to assess autophagic flux is to use bafilomycin or chloroquine in basal conditions and in nutrient starved conditions to block autophagosome degradation and assess intensity of LC3-I/LC3-II. The rescue experiment with pAC is interesting but this western blot is very problematic. All samples should be run in the same blot to get a better estimate of LC3-I and LC3-II in comparison to A375 and A375-null cells. Furthermore, this WB seems to duplicate actin from Figure 1C but LC3 seems to be from a different blot? Finally, the authors should indicate the conditions for the immunofluorescence in Figure 1h (basal or starved). We recommend staining for LC3 instead of/in addition to lysotracker to check for LC3 puncta. We recommend doing this experiment in basal and nutrient starved conditions +/- chloroquine or bafilomycin. This will be a better indicator of autophagic flux in nutrient starved conditions.

We added an high content confocal screening following reviewer suggestions. We measured LC3-RFP puncta (autolysosomes) Lines 118-125 “Then, we analysed autophagic flux by transfecting A375 and A375 AC-null cells with pLC3-EGFP-mRFP. The analysis was performed by counting the number of LC3-RFP puncta (autolysosomes) in basal and starving condition. To better evidence differences in terms of autophagosome accumulation, we also added chloroquine (CHQ) 1 h prior fixation, a potent inhibitor of autolysosome formation. The screening revealed that A375 AC-null cells  have less LC3-RFP puncta  compared to A375, interestingly, we observed no variation of LC3 puncta when CHQ is added to AC-null cells (Figure 2f, g)

Figure 2 suggests AC ablation promotes apoptosis after nutrient deprivation. For any cell-death experiment by flow, controls in basal conditions should be provided for all cell lines. The authors focus on the reduction in apoptotic cells with pAC rescue (24.8 to 9%) but the % of viable cells is only 10-20% in all conditions. These results don’t match with the data presented in Figure 1 suggesting that nutrient starvation does not impact cell viability. Controls in basal conditions would help to assess the gating strategy and to differentiate if those PI+ cells are really necrotic cells or artifacts.

We apologise for the lack of clarity. We tested two different types of starving: the first one, described in figure 1, was performed using a glucose free media. The second one was performed using HBSS media, without any kind of nutrients (no sera, no glucose, no amino acids, no vitamins). Lines 102-105 “Cells were exposed to glucose free medium for 24 h, and ASAH1 mRNA levels were measured by q RT-PCR. The results show that glucose deprivation caused a significant increase in AC transcription (Figure 1b), this was also confirmed by immunocytochemistry assays (figure 1c-d). Interestingly, glucose starvation did not affected A375 cell viability (Figure 1e).”.  and lines 140-142 “We starved A375 and A375 AC-null cells, transfected or not with pAC, using HBSS medium for 2 h. HBSS did not contain glucose, sera, amino acids, vitamins and other micronutrients”.

We changed the dot-plot gate to clarify the apoptosis analysis, definition of Q4 is added. Lines 145-147 were added “we discriminated living cells (Caspase 3-7 negative /PI negative), dead cells (Caspase 3-7 negative /PI positive), apoptotic (Caspase 3-7positive /PI negative), and necrotic (Caspase 3-7 positive/PI positive).” We also added A375 negative control in Figure 3.

In figure 3, the authors show a decrease in MITF and PGC1 expression in AC null cells which would be suggestive of decreased mitochondrial biogenesis. Next, they evaluate mitochondrial mass with mitotracker green and mitochondrial polarization with mitotracker red. The authors quantify mito green by FACS and they show no differences between cell lines. Nevertheless, the IF shows higher staining in the null cells. WB for mitochondrial markers such as Tom20 or others are encouraged. Furthermore, based on the IF results, the authors suggest mitochondrial morphology is altered in AC KO. This is interesting but a quantification of % of fragmented vs elongated mitochondria would be required. Also, how do the authors interpret an increase in MFN1 and DRP1 expression by RT-PCR with the differences in mitochondrial elongation.

We thank the reviewer for its suggestions. We corrected the text and divided the mitochondria analysis in two parts: the first shows mRNA expression and total mitochondria measurements by flow cytometry (lines 177-182). The second, shown in the new figure 5,  analyses structure and mitochondrial membrane potential using high-content confocal microscopy analysis, lines 183-204. Moreover we provide an high content  texture analysis that gives hints on how mitochondria change their organization and shape in both cell lines. MFN1 and DRP1 differential expression are now discussed in lines 183-189 “Then, we probed expression levels of MFN-1 and DRP-1 genes. These genes, among others, regulate mitochondrial fusion (MFN-1) and fission (DRP-1), respectively. As shown in Figure 4e-f, we found MFN-1 mRNA expression up-regulated in A375 AC-null cells, as we observed an increase in DRP-1 expression, this was not statistically relevant. Collectively, qRT-PCRs indicate a derangement of mitochondrial network dynamics in A375 AC-null cells. Starting from this finding, we probed AC-null cells for mitochondria integrity and membrane polarization, using high content confocal microscopy screening”.

Reviewer 3 Report

The manuscript “Ablation of acid ceramidase results in impaired autophagy and mitochondria depolarization in melanoma cells” by Lai et al, identifies key components of acid ceramidase (AC) activity in regulating melanoma chemoresistance. The experiments and analyses performed are of good quality. The observations should be of importance in the prognoses of malignant melanoma. Therefore, the presented results stand to serve as an important template for future investigations and translational science. On that regard, I urge the authors to respond to the following questions/issues.

  1. One major point is that only A375 cells were used to validate the hypothesis. When the authors engaged SK-MEL-28 cells using RNAi, no difference in LC3 II levels were observed – leading the authors to conclude that even low levels of AC might be adequate to initiate/maintain autophagy. If that is true, then it poses a severe hurdle against clinical efficacy of AC inhibitors. Can the authors ascertain whether RNAi in a few other melanoma cell lines will yield the same results? It is risky to conclude from a single cell line, given the heterogeneity of cancer as a disease.
  2. Are the A375-AC-null cells more chemoresistant than A375 -AC-sufficient cells (as the authors correctly point out that active autophagy favors chemoresistance)? A comparison of EC50 of those two cell lines for dacarbazine, cisplatin, doxorubicin, etc. can easily provide answers to this.
  3. In publicly available melanoma datasets, how does AC expression correlate with overall mortality? Is there a sex-specificity in that?

Author Response

Manuscript ID: ijms-1094810

Point by point replies to reviewer 3

The manuscript “Ablation of acid ceramidase results in impaired autophagy and mitochondria depolarization in melanoma cells” by Lai et al, identifies key components of acid ceramidase (AC) activity in regulating melanoma chemoresistance. The experiments and analyses performed are of good quality. The observations should be of importance in the prognoses of malignant melanoma. Therefore, the presented results stand to serve as an important template for future investigations and translational science. On that regard, I urge the authors to respond to the following questions/issues.

We thank the reviewer for its opinion on our work

  1. One major point is that only A375 cells were used to validate the hypothesis. When the authors engaged SK-MEL-28 cells using RNAi, no difference in LC3 II levels were observed – leading the authors to conclude that even low levels of AC might be adequate to initiate/maintain autophagy. If that is true, then it poses a severe hurdle against clinical efficacy of AC inhibitors. Can the authors ascertain whether RNAi in a few other melanoma cell lines will yield the same results? It is risky to conclude from a single cell line, given the heterogeneity of cancer as a disease.

We agree with reviewer, AC quantification can be a parameter to taken into account when AC inhibitors are proposed as adjuvant therapy.

We added other melanoma cell lines (M14 and SK-Mel28) which completely differs to A375 in terms of mutations, phenotype and AC expression levels (see Figure 1a). AC-overexpressing cells (M14 and SK-Mel 28) did not modulate autophagy even when transfected with AC siRNA. Probably because we did not totally suppress AC transcription, as shown in in figure S1b-c. This result is now discussed in lines 125-133 “These results, together with those illustrated in Figure 2a, indicate that AC deletion decreases autolysosomes. To extend this finding to melanoma cells that accumulate AC (Figure 1a), we transfected AC-siRNA into M14 and SK-MEL-28, melanoma cell lines known to express high levels of AC [27]. As shown in Supplementary Figure 1b-c, siRNA transfection decreased but did not suppress the levels of AC transcripts. Western blot analysis shown in Supplementary Figure 1c revealed no differences on LC3 II content in both M14 and SK-MEL-28 cells transfected with AC-siRNA. These findings indicate that AC expression, even at low levels, might be sufficient to maintain active autophagy.”.

  1. Are the A375-AC-null cells more chemoresistant than A375 -AC-sufficient cells (as the authors correctly point out that active autophagy favors chemoresistance)? A comparison of EC50 of those two cell lines for dacarbazine, cisplatin, doxorubicin, etc. can easily provide answers to this.

No, A375 AC-null cells re-sensitize to chemotherapy, EC50 and vitality assays of both cell lines are described in our previous work cited in the text as [16]. Lines 321-322 were added “It is well established that AC inhibitors reduce resistance to chemotherapy [5] and radiotherapy of melanoma and several other types of cancer [14,16].”

  1. In publicly available melanoma datasets, how does AC expression correlate with overall mortality? Is there a sex-specificity in that?

This is a very interesting question, unfortunately as far as we know there are no available datasets that  can be used to correlate AC expression with overall mortality in melanoma, but in the recent 10 years several correlations between AC and melanoma plasticity have been observed by many independent research groups.

Round 2

Reviewer 1 Report

The authors have answered all the questions, and thanks to their efforts.

Reviewer 2 Report

The authors have replied to all major and minor comments and have significantly improved the quality of the manuscript. I recommend the manuscript for publication.